# Target Recognition– and HCR Amplification–Induced In Situ Electrochemical Signal Probe Synthesis Strategy for Trace ctDNA Analysis

**DOI:** 10.3390/bios12110989

**Published:** 2022-11-08

**Authors:** Aiting Cai, Luxia Yang, Xiaoxia Kang, Jinxia Liu, Feng Wang, Haiwei Ji, Qi Wang, Mingmin Wu, Guo Li, Xiaobo Zhou, Yuling Qin, Li Wu

**Affiliations:** 1School of Public Health, Nantong University, No.9 Seyuan Road, Nantong 226019, China; 2Department of Laboratory Medicine, Affiliated Hospital of Nantong University, Medical School of Nantong University, Nantong 226001, China

**Keywords:** DNA metallization, ctDNA, HCR, Exo Ι, E-DNA sensor

## Abstract

An electrochemical-DNA (E-DNA) sensor was constructed by using DNA metallization to produce an electrochemical signal reporter in situ and hybridization chain reaction (HCR) as signal amplification strategy. The cyclic voltammetry (CV) technique was used to characterize the electrochemical solid-state Ag/AgCl process. Moreover, the enzyme cleavage technique was introduced to reduce background signals and further improve recognition accuracy. On the basis of these techniques, the as-prepared E-DNA sensor exhibited superior sensing performance for trace ctDNA analysis with a detection range of 0.5 fM to 10 pM and a detection limit of 7 aM. The proposed E-DNA sensor also displayed excellent selectivity, satisfied repeatability and stability, and had good recovery, all of which supports its potential applications for future clinical sample analysis.

## 1. Introduction

Liquid biopsy has long been accepted as an advanced technique to obtain testing samples from body fluids because of its convenience and low invasiveness [1,2,3]; it is widely utilized in early cancer detection, monitoring of micrometastatic disease, treatment aids, and prognostic diagnostics [4,5]. The level of circulating tumour DNA (ctDNA) in the blood has proved to be closely correlated with tumour type, progression, burden, growth, and treatment [6]; it thereby seems to be an efficient molecular marker for cancer diagnostics and treatment [7,8]. However, the analysis of ctDNA in body fluid samples is limited by its particularly low abundance [9] and short half-life (usually < 2 h) [10], demanding a highly sensitive and fast detection approach. Although the digital polymerase chain reaction (PCR) [11], quantitative PCR, and next-generation sequencing techniques [12] have been developed for ctDNA analysis, these technologies still suffer from the shortcomings of complex operation, expensive cost, cumbersome design, time consumption, and requirement of higher experimental manipulation skills [13,14,15,16]. Therefore, it is urgent to develop a highly sensitive, convenient, and low-cost ctDNA detection technology.

Electrochemical DNA (E-DNA) sensors exhibit natural advantages for ctDNA detection due to their convenient construction, low cost, time saving, and high specificity. In addition, these sensors are easy to miniaturize into portable instruments for point-of-care testing [17,18,19]. Isothermal amplification technology refers to the rapid amplification of nucleic acids by adding different enzymes and specific primers with different functions to the reaction at a constant temperature. Because of the advantages of E-DNA sensors, isothermal amplification technology [20,21] has been successfully integrated to improve sensing performance owing to its flexible structure design, mild and easy-to-control reaction conditions, and low experimental cost [22,23,24,25]. However, the existing technology is far from meeting the requirements of ctDNA detection at the clinical level, as the sensitivity and accuracy of most of the presented methods still need to be improved.

In this study, taking KRAS gene as a ctDNA model, an E-DNA sensor was constructed for ctDNA sensing by fully integrating the advantages of DNA metallization [26,27], HCR, and Exonuclease Ι (Exo Ι) cleavage technique, named as an EC-HCR sensor [28]. DNA metallization strategy can produce a portion of the metal tracer and then the largely amplifying electrochemical signal. Moreover, the detection signals can be further amplified on the basis of the HCR for DNA. Further, Exo Ι can diminish the background signal and enhance the signal-to-noise ratio. The process of sensor construction is depicted in Figure 1. Primarily, the capture probes were immobilized onto the bare electrode surface through the interaction of gold–thiol. Then 6-mercapto-1-hexanol (MCH) was introduced to block the left sites on the electrode surface, thereby reducing false positive signals. The ctDNA KRAS mutation 134A allele was used as the model target. Exo Ι was able to digest single-strand DNA from a 3′→5′ direction, cutting off the capture probes that were not hybridized to duplex with the target. The initiator, connecting the target and hairpin H1, initiated the HCR [29] to generate a double-strand nanostructure with repetitive units. Finally, silver ions were introduced on the electrode surface by absorbing on phosphate groups and nitrogen atoms in the nucleic acid bases. Subsequent reduction treatment realized DNA metallization, producing silver nanoparticles in situ along the abovementioned double-strand structure, serving as the electrochemical probes for KRAS gene detection. A single ctDNA recognition event initiated the exponential production of signal reporters, thus significantly amplifying the output electrochemical signal. Taking the unique electrochemical advantage of Ag-AgCl solid-phase transformation [27], an electrochemical signal that was proportional to ctDNA concentration was measured. Consequently, the as-developed ctDNA sensor exhibited a good linearity ranging from 0.5 fM to 10 pM with a lower limit of detection (LOD) of 7 aM. The proposed strategy is promising to be expanded for efficient analysis of other ctDNA, thus providing a new detection strategy for the early diagnosis of cancer. To the best of our knowledge, this study marks the first time that the concept of target-triggered in situ production of electrochemical signal probes, the DNA chain amplification reaction, and the enzymatic degradation of DNA have been flexibly integrated in one system for improving the sensing performance of ctDNA analysis.

## 2. Materials and Methods

### 2.1. Chemical Reagents and Materials

Acrylamide, N,N′-Methylenebisacrylamide, Tris Base, and NaCl were obtained from Shanghai Aladdin Bio-Chem Technology Co., Ltd. Ammonium persulfate, HCHO, K_3_Fe(CN)_6_, K_4_Fe(CN)_6_, and N,N,N′,N′-Tetramethylethylenediamine (TEMED) were obtained from Shanghai Macklin Biochemical Co., Ltd. HNO_3_, AgNO_3_ and HCl were obtained from Shanghai Lingfeng Chemical Reagents Co., Ltd. Orthoboric acid was prepared from Sigma-Aldrich (Shanghai) Trading Co, Ltd. MCH was obtained from Shanghai Yien Chemical Technology Co., Ltd. NaNO_3_, Mg(NO_3_)_2_·6 H_2_O and H_2_SO_4_ were obtained from Sinopharm Chemical Reagent Co., Ltd. NaCO_3_ was obtained from Shanghai Runjie Chemical Reagent Co., Ltd.

All the sequences used in this study were synthesized and offered by Sangon Biotech Co., Ltd. (Shanghai, China) (Appendix A). Exo Ι was purchased from Yuanye Bio-Technology Co., Ltd. (Shanghai, China).

### 2.2. Apparatus

The measurements of cyclic voltammetry (CV) and electrochemical impedance spectroscopy (EIS) were performed on an Autolab PGSTAT 302 N electrochemical station (Metrohm Technology Co., Ltd., Herisau, Switzerland). A polyacrylamide gel electrophoresis experiment was used on a DYCZ-25 D electrophoresis apparatus (Beijing 61 Biological Technology Co., Ltd., Beijing, China) and imaged on a Tanon 2500 BR Multifunctional gel imaging analyzer (Tanon, Shanghai, China). The determination of the nucleic acids’ concentration was conducted with a UV-1900 spectrophotometer (Shimadzu, Japan). Chronocoulometric experiments were conducted at a CHI 660 E electrochemical workstation (Shanghai Chenhua Instruments Co., Ltd., Shanghai, China).

### 2.3. Fabrication of E-DNA Sensor

An Au electrode was polished by 0.3 μm and 0.05 μm aluminum oxide suspension and cleaned via muti-step electrochemical cyclic voltammetry cleaning steps of four different solutions (0.5 M NaOH, 0.5 M H_2_SO_4_, 0.01 M KCl/0.1 M H_2_SO_4_, and 0.05 M H_2_SO_4_).

First, the Au electrode was dropped by a 6 μL capture probe (0.05 μM) and incubated for 1.5 h at room temperature. Then, 6 µL MCH (1 mM) was dripped onto the Au electrode and incubated for 20 min to fill the blank area and prevent unspecific absorption. In the following step, 6 µL target solution was dropped on the electrode for 1 h. Then the electrode was rinsed with ultrapure water. Furthermore, the modified electrode was incubated with Exo Ι (0.1 U/mL) for 10 min at 37 °C to cleavage the capture probes, which were not hybridized to duplex with the target. Then the electrode was again rinsed with ultrapure water. 6 µL initiator solution (0.5 µM) was put on the electrode for 1 h to leave the trigger domain to trigger subsequent aggregation between hairpins H1 and H2. After the electrode was rinsed again with ultrapure water, hairpins H1 and H2 were annealed in their own tubes by heating to 95 °C for 5 min, then slowly cooled down to room temperature. Afterward, we mixed them to 1.5 µM by DNA hybridization buffer for further use. Finally, the mixed solution (1.5 µM) was applied to the electrode incubation for 1.5 h to generate HCR. Then the electrode was rinsed once more with ultrapure water to proceed with the later operation. The DNA hybridization buffer and HCR buffer contain 50 mM Mg(NO_3_)_2_·6H_2_O and 100 mM NaNO_3_ in 20 mM HEPES (pH 7.4). The HCR solution contains 0.5 µM initiator solution and 1.5 µM hairpin H1 and H2 mixture.

### 2.4. Electrochemical Measurements

The three-electrode system was applied in the electrochemical measurements used, including Au electrode as the work electrode, Ag/AgCl (3 M KCl) as the reference electrode, and platinum wire as the auxiliary electrode. Cyclic voltammetry (CV) was carried out in 10 mM K_3_[Fe(CN)_6_] containing 1 M KCl, at a 0.1 V/s of scan rate and ranging from −0.2 V to 0.7 V. Electrochemical impedance spectroscopy (EIS) was conducted in 10 mM K_3_[Fe(CN)_6_]/K_4_[Fe(CN)_6_] containing 1 M KCl. The parameters for the EIS were set as 0.01 V of the amplitude and 0.01 Hz to 1 × 10^5^ Hz of the frequency range. The Chronocoulometric experiment was operated in 10 mM Tris-HCl (pH 7.4) containing 50 μM RuHeX. 

For the DNA metallization, the modified electrode was immersed in AgNO_3_ solution (100 µM) for 1 h and the electrode was then rinsed with ultrapure water. Subsequently, the electrode was soaked in the freshly prepared NaBH_4_ (10 mM) for 10 min to reduce the Ag ions to Ag on the DNA strand. Then the electrode was rinsed with ultrapure water. The electrode was placed in 0.1 M NaCl containing 0.1 M NaNO_3_ as electrolyte and measured ranging from −0.2 to 0.4 V.

### 2.5. Polyacrylamide Gel Electrophoresis (PAGE)

5 μL HCR product with 1 μL loading buffer was analyzed in 1 × TBE buffer by 15% PAGE (30% Poly-AcrylaMide, 5 × Tris-Borate buffer (TBE), 10% Ammonium persulfate (APS), TEMED). The conditions of PAGE were 100 V of voltage and 40 min of time. Gel electrophoresis was stained using the silver-stained method (10% ethanol, 0.7% HNO_3_, 0.2% AgNO_3_, developer) and scanned in a gel imaging analyzer.

## 3. Results and Discussion

### 3.1. Feasibility Study of Exo Ι and HCR Assay (In Liquid Phase)

The single-chain cleavage function of Exo Ι and the formation HCR reaction were verified by electrophoresis (PAGE). The results showed that lanes 1, 3, 4, 5, and 6 corresponded to the five components (capture probe, target, initiator, hairpin H1, and hairpin H2) participating in HCR reaction, respectively (Figure 1, channel information in Appendix A), while the migration rate was consistent with their own base numbers. Exo Ι, as a specific nuclease cutting ssDNA from 3′ to 5′, was used to eliminate background signals by removing the unreacted capture probes. Once the capture probe was combined with a complementary target, the hybridization led to the closure of the 3′ end, disturbing the cleavage site of Exo Ι. When the capture probe mixed and incubated with Exo Ι, the single-stranded probe was digested and no obvious electrophoresis band was observed (lane 2). On the contrary, when the capture probe and target bonded bound to form a double strand and then incubated together with Exo Ι, the double-strand product could not be digested and a bright band still existed (lane 7, 8). Therefore, Exo Ι showed exonuclease activity to degrade the single-stranded DNA from 3′ to 5′ with no activity on the double-stranded DNA. When the target and initiator were mixed and incubated, a band with a slightly larger relative molecular weight than the initiator appeared, while some unreacted target remained (lane 9). When the initiator and hairpin H1 were mixed and incubated, a band with a very large relative molecular weight appeared, while some unreacted hairpin H1 remained (lane 10). By contrast, when the two hairpins were mixed, the binding reaction did not occur and no new bands appeared, as there was no initiator. Because the two sequences had exactly the same relative molecular weight, only one band appeared (lane 11). However, when the initiator was present, the binding reaction between hairpin H1 and H2 was able to start, the new band appeared, and the unreacted hairpins H1 and H2 were left (lane 13). When the capture probe, target, and initiator were incubated, the results showed two bands, one representing the combination of the three, and the other representing the hybridization of the capture probe and target (lane 12). Once the capture probe, target, initiator, hairpin H1, and hairpin H2 were mixed and cultivated together, a high molecular weight product appeared and the unreacted hairpins H1 and H2 were left (lane 14). The above results confirmed that the HCR reaction would occur between the five nucleic acids. On the other hand, the feasibility of the above constructed E-DNA sensor was also verified preliminarily. The stepwise construction of the E-DNA sensor was monitored and confirmed by traditional cyclic voltammetry (CV) and EIS methods, the details of which are shown in the supporting information (Appendix A).

### 3.2. Verification of DNA Metallization

DNA metallization is the way to realize free-label electrochemical signal amplification in our systems. The dsDNA product with negative charge yielded by HCR can attract silver ions in the solution, while Ag^+^ is reduced to a metal cluster in situ. The implementation of this method is restricted to a negatively charged ribose phosphate backbone, which possesses excellent selectivity and avoids interference caused by nonspecific signals, thus improving accuracy. Figure 2a illustrates that the transformation between silver nanoparticles and solid AgCl is reversible, while sharp sliver oxidative peak (potential at 0.15 V) and reductive peak (potential at 0.04 V) demonstrated the successful accomplishment of DNA metallization. After metallization of the HCR product, the redox potential of [Fe(CN)_6_]^3−/4−^ at the electrode was altered and the current was increased, as shown in Figure 2b. The result revealed a potential difference of 0.1 V from 0.4 V to 0.3 V (curve a to curve b). Because there was still 1 M KCl in the CV test solution, the Ag redox peak appeared in the CV plot. The Ag redox potential was changed by the influence of [Fe(CN)_6_]^3−/4−^ (the new peak appeared in curve b). Additionally, the decrease in the EIS Ret value (from over 2000 Ω to about 70 Ω) can be observed clearly (Figure 2c). Consequently, the above experiments indicated that DNA metallization caused an enlargement of the electron transfer rate on the electrode interface and an improvement in conductivity. The morphology and distribution of the Ag nanoparticles formed by the DNA metallization were characterized by transmission electron microscopy (TEM) (Appendix A). The Ag nanoparticles showed a uniform distribution and a spherical shape.

### 3.3. Exo Ι—Assisted Background Suppression Strategy

The signal-to-noise ratio (SNR) is one of the key indicators to evaluate the sensing performance of the established sensor. Although DNA metallization is an excellent technique to amplify electrochemical signals, it is often subject to the factor of excessive background signals. Exo Ι can digest a capture probe from 3′ to 5′ direction, while the cutting products of the capture probe would eventually be removed from the surface by a buffer. From Appendix A, it can be seen that the Exo Ι processing strategy was favourable to decrease the current peak and suppress the background signal. The intensity value of the peak (Appendix A) obtained with Exo Ι treatment decreased 94% in contrast to the result without Exo Ι treatment, confirming the feasibility of the scheme to eliminate the background signal. 

### 3.4. Signal Amplification of HCR

The extended duplex DNA via HCR reaction created more sites for DNA metallization, which amplified the signal. Under the same target concentration condition (1 nM), the current without HCR amplification was about 1.2 µA while the current with HCR treatment was about 4.9 µA (Figure 3a). When the target was absent and all the other components were present, the capture probe was completely cut by Exo Ι without a current signal. The current intensity greatly increased after the HCR to about 5 times higher than that without HCR, confirming the absolute feasibility of the amplification strategy (Figure 3b).

### 3.5. Performance of the Constructed EC-HCR Sensor

Using the optimized parameters given in Appendix A, the detection performance of the above-constructed sensors for ctDNA was further examined. Sensitivity was reflected by the detection range and the detection limit. As shown in Figure 4a, the strength of the electrochemical signal gradually increased with the increase in ctDNA concentration. However, when the concentration increased further, the current value approached a plateau (Figure 4b). The inset showed close linear correlation between the peak current and concentration at the target concentration from 0.5 fM to 10 pM, and the correlation coefficient (R^2^) was 0.992. The linear correlation curve was described as y = 1.0399 x + 0.5314. According to the calculation basis (3σ rule), the final LOD was calculated to be as low as 7 aM, which was lower than in previous studies, indicating that the proposed sensor exhibited excellent sensitivity (Appendix A). The superior performance can be attributed to the efficient HCR amplification and the unique way in which the electrochemical signals are generated.

In order to examine the selectivity of the prepared E-DNA sensor, single-base mismatched sequences, double-base mismatched sequences, and three-base mismatched sequences were chosen for the following test. Under the same conditions, the current value obtained from the target recognition was significantly higher than that from mismatch sequences recognition (Figure 4c), indicating that the E-DNA sensor possessed excellent selectivity and could distinguish target and non-target sequences clearly. On the basis of the amplification effect of the DNA metallization and HCR, the individual molecular recognition differences between single-base mutation can be converted into large different electrochemical signals through the abovementioned amplification strategies, achieving highly selective single-base mismatch discrimination. Five parallel electrochemical sensors in Figure 4d were constructed, and the relative standard deviation (RSD) was 5.60%, revealing the accuracy of the sensor. The stability showed that the current strength could be preserved 84% after 7 days. The results show that the sensor has good stability (Appendix A).

### 3.6. Real Sample Testing

The recovery experiment can verify the potential of the constructed sensor in clinical applications. Human serum from normal volunteers was drawn using the approved blood drawing technique. The complex matrix solution was yielded by diluting the human serum 100 times using a buffer. The patients gave informed consent, and the study had the approval of the ethical committee of Nantong University Affiliated Hospital. The target standard solution was added as a reference for recovery experiments. The target standard solution used for analysis in the sample was commercial target single-chain. On the basis of the linear equation, the concentrations of target in the real samples were calculated and the satisfactory recovery rate (95.7–106.0%) and the RSD (3.8–6.5%) were achieved (Appendix A), exhibiting a huge potential for this strategy in future clinical sample analysis.

## 4. Conclusions

In summary, an EC-HCR sensor was proposed by using target recognition–induced in situ generation of silver nanoparticles as electrochemical probes for ctDNA KRAS gene detection, while HCR was used to amplify the output signal, and enzyme digestion technique was applied to suppress the background signal. Silver metallization can be used to indicate DNA composition like traditional electrophoresis, providing a specific-target nucleic acid–dependent electrochemical signal. On the basis of the above design, the as-prepared electrochemical DNA sensors exhibited wider linear detection range and lower detection limit. They also displayed outstanding selectivity, repeatability, and stability. This recovery experiment has huge potential for the future analysis of clinical samples.

## Data Availability

The data presented in this study are available on request from the corresponding author.

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
