# Peer review of "Target Recognition– and HCR Amplification–Induced In Situ Electrochemical Signal Probe Synthesis Strategy for Trace ctDNA Analysis"

_biosensors, 2022, doi:10.3390/bios12110989_

Round 1

Author Response

Reviewer #1:

The article focuses on developing an electrochemical sensor for sensing single-stranded DNA using HCR and DNA metallization as a signal amplification strategy. Authors implemented Exo I digestion to enhance signal-to-noise ratio. The approach established in the study is interesting, however, the authors should clarify or modify the article as per the following suggestions comments.

Abstract:

Q1): What electrochemical reaction is used in the proposed sensor? Also, it should be mentioned clearly throughout the article where it is appropriate.

Response: The electrochemical reaction in the proposed sensor refers to the solid-state Ag/AgCl reaction. Performing cyclic voltammetry measurements at enough electrolyte concentration, two well-separated sharp current peaks are observed, which are attributed to the oxidation of Ag to AgCl and reduction of AgCl back to Ag, respectively. The charge efficiency for the oxidation and subsequent reduction is near 100%, suggesting reversible conversion of the AgNPs to the silver halide phase during the anodic scan and subsequent conversion back to Ag during the cathodic scan (DOI: 10.1002/adfm.201303818).

Introduction:

Q2): Page2, line number 51: what is old electrode?

Response: The old electrode refers to the bare Au electrode. We have double-checked and revised the manuscript (please see the Page 2, line number 55 in the revised manuscript).

Q3): Page2, line number 52: what is MCH?

Response: MCH refers to 6-mercapto-1-hexanol, which contains -SH groups that can form Au-S bonds with gold electrode to avoid non-specific absorption. We have supplemented the full name of MCH in the manuscript (please see Page 2, line number 56 in the revised manuscript).

Q4): Page2, line number 70: what is DNA chain amplification reaction?

Response: DNA chain amplification reaction in page 2 represents the hybridization chain reaction (HCR). Hybridization chain reaction (HCR) is a typical enzyme-free, self-catalytic strategy to construct DNA nanostructure for DNA detection with PCR-like sensitivity, in which the hybridization event is initiated by initiator sequence and ultimately forms DNA double-stranded by continuous and ordered polymerization of complementary hairpin sequence.

Q5): Page2, lines number 47-72: the whole para deals with a summary of the current article. The introduction section lacks a statement of the problem regards to novelty of the sensor design proposed in this article.

Response: Many thanks for the reviewer’s valuable advice. We have supplemented the innovation of the sensor design proposed in the introduction (please see the Page 2, line number 50-54 in the revised manuscript).

Methods:

Q6): Section 2.3 The composition of Initiator solution, DNA hybrid buffer, and HCR buffer should be clearly stated. HCR reaction conditions should be elaborated

Response: The initiator could initiate the HCR under the participation of hairpin H1 and H2 to generate a double-strand nanostructure with repetitive units. The sequence of initiators in this article is AGC TCC AAC AGT CTA GGA TTC GGC GTG GGT TAA. The concentration of the initiator solution is 0.5 µM. The DNA hybrid buffer and HCR buffer contain 50 mM Mg(NO3)2·6 H2O and 100 mM NaNO3 in 20 mM HEPES (pH 7.4). The HCR solution contains 0.5 µM initiator and 1.5 µM hairpin H1 and H2 mixtures. We have double-checked and revised the manuscript (please see Page 3, line number 120-122 in the revised manuscript).

Q7): Section 2.4 newly prepared NaBH4 can be rewritten as freshly prepared NaBH4

Response: We appreciate the reviewer for the valuable suggestion. We have double-checked and revised the manuscript (please see Page 4, line number 134 in the revised manuscript).

Q8): Section 2.5 should be rewritten with standard phrases used for PAGE

Response: We appreciate the reviewer for the valuable suggestion. We have double-checked and revised the manuscript (please see the Page 4, line number 139-143 in the revised manuscript).

Result and discussion:

Q9): Section 3.1 What is single chain cleavage?

Response: The single chain cleavage refers that single chain (ssDNA) can be cleaved by Exo Ι. Exo Ι possesses exonuclease activity to cleavage single stranded DNA from 3' to 5' while no activity on double-stranded DNA (DOI: 10.1016/j.bios.2018.09.081).

Q10): Instead of “Cut thoroughly” use “digested”

Response: We appreciate the reviewer for the valuable suggestion. We have double-checked and revised the manuscript (please see the Page 4, line number 154 in the revised manuscript).

Q11): Line number 152: what does it mean “double stranded” in that sentence?

Response: The “double stranded” in our manuscript should be revised to capture probe. We have double-checked and revised the manuscript (please see the Page 4, line number 155-157 in the revised manuscript).

Q12): Fig.1 Lane number 12 is not matching the description in the section 3.1

Response: We appreciate the reviewer’s kind reminder. We have double-checked and revised in section 3.1 (please see the Page 4, line number 170 in the revised manuscript).

Q13): Section 3.2 Line number 183: What is “negative DNA structure”?

Response: We appreciate the reviewer for the valuable suggestion. Negative DNA structure represents dsDNA product with negative charge yielding by HCR. Since DNA possesses rich phosphate groups, it can bind with silver ions by electrostatic attraction.

Q14): Line number 187: which diagram?

Response: This description here corresponds to Figure 2a.

Q15): Line number 198: how does metallization improve the conductivity? How it is relevant or useful to increase the sensor performance?

Response: The electrochemical properties of silver deposition along the DNA template confined at electrode surface was investigated by cyclic voltammetry and EIS, respectively (Figure 2b, c). The increase of peak current in cyclic voltammograms and the decrease of Ret in EIS spectra demonstrate that a complete metallization of DNA has severely enhanced the conductance properties of DNA. These results are reasonable and in accordance with previous work, pointing out that DNA-templated metal deposition is an attractive approach to improve the conductivity of electrode interfaces and increase the sensor performance. (DOI: 10.1021/nl035124z, DOI:10.1016/S1388-2481(02)00542-8).

Q16): Figure4d: what the error bar is representing?

Response: The error bar represents the electrochemical sensor was constructed three times on the same electrode under the same experimental conditions.

Reviewer 2 Report

The manuscript by Cai et al., described an electrochemical sensor for ctDNA KRAS gene based on DNA metallization and HCR. The sensor allowed the satisfactory analytical performance for the detection of ctDNA KRAS gene, and it was successfully applied for the clinical sample analysis. It is recommended to be published after a minor revision. Comments and suggestion are listed below.

1) The main content in the abstract should include important conclusions and shining results of this study, e.g., the representative data, the underlying mechanisms and the future enlightenment.

2) How about the stability of the sensor?

3) In Figure S4 (e), the incubation time shows relatively weak influence on the current. Why chose 1.5 hour as the optimal value?

4) The linear range of the sensor is 0.5 fM-10 pM, and the detection limit is 7aM. The calculation details should be provided.

5)  The English writing needs improvement.

Author Response

(The authors gave the same response as above.)

Reviewer 3 Report

The article is devoted to the development of a highly sensitive electrochemical biosensor for DNA detection using additional target treatment with Exonuclease 1 and subsequent HCR to amplify the detection signal. In the paper showed very promising results. Extremely high sensitivity of detection 1800 molecules in sample (0.5 fM, 6 ul of sample) was shown in the study. Moreover, using the three sigma rule they are proposing the 7 aM concentration (25 molecules) detection limit. However, many issues need to be resolved before publication. The manuscript needs major revision and after corrections re-revised.

Major problems are following:

1)      There are a number of publication on electrochemical DNA biosensor with HCR amplification (for example, https://doi.org/10.1016/j.bios.2018.04.046, https://doi.org/10.1155/2017/4571614, https://doi.org/10.1016/j.talanta.2017.11.036) and HCR-Ag optical  (https://doi.org/10.1039/D0RA04202A) or electrochemical (https://doi.org/10.1016/S1388-2481(02)00542-8) sensors. The novelty of the paper should be more clearly indicated.

2)      In the manuscript, the authors propose to use the KRAS gene ctDNA as a target. Typical ctDNA size is 166 base pairs (doi: 10.1073/pnas.1500076112). The authors propose to use short 19 nt fragments of model single stranded DNA. There are a number of problems for the use of ctDNA. It should be discussed in the manuscript.

(a) It is not clear whether the KRAS gene fragment is present as ctDNA in the blood?

(b) How to denature double stranded DNA and remove the second strand to prevent renaturation?

(c) What is the position of the selected target fragment in ctDNA and the location of the mutation? This is important for Exo 1 treatments.

3)      How the sequence of target in 134A allele was selected? What is the origin of the choice of length and sequence for the target, the mismatches, and all the components of the system under study?

4)      In section 3.3. a significant increase in current was observed after HCR. A reference experiment without a template but in the presence of all other components is required to validate the signal specifity.

5)      Then “Then, 6 µ L MCH (1 mM) was dripped onto the Au electrode and incubated for 20 min to fill the blank area and prevent unspecific absorption.” The prevention of non-specific absorption was not shown.

6)      “Taking the unique electrochemical advantage of Ag-AgCl solid-phase transformation [27], electrochemical signal that was proportional to ctDNA concentration could be measured.” I cannot fully agree with this statement. The signal is proportional not only to the ctDNA concentration, but also to the length of the amplified HCR chain.

7)      The origin of the extremely high selectivity (single mismatch) presented in Figure 4(c) needs to be discussed.

8)      Additional information on testing of real samples must be provided (section 3.6).

(a)    “The complex matrix solution was prepared by diluting human serum (from health volunteers) 100 times.” This sentence is not clear.

(b)   This Table S2 is not clear.

(c)    How many samples/donors was analyzed? What volume of serum was prepered and taken for analysis?

(d)   The concentration of ctDNA was not measured by alternative approach in the serum.

(e)    What does mean recovery experiment?

(f)     Do you have permeation from ethical committee for experiments with volunteers and serum?

9)      “The recovery experiment realized in human serum sample provided not only a brand-new platform for early diagnosis of cancer, but also huge potential for the future analysis of other clinical samples.” To make this conclusion authors should analyze typical concentration of ctDNA at different cancer stages and compare them with the data obtained.

Minor issues:

1)      In general, it is not correct to compare concentration limits of assays at different volumes used in the assay (Table S1). The molar quantity of samples is more appropriate.

2)      Authors uses MCH (is it mercaptohexanol?) “to block the left sites on the electrode surface, thereby reducing the false positive signals”. Why don't you use a mixture of MCH with oligonucleotides to control the surface density of capture probes and fulfilling the surface?

3)      Authors claim that they “incubated for 1.5 h at room temperature to form self-assembled monolayers.” How did you confirm the monolayer formation?

4)       “After the electrode was rinsed with ultrapure water, hairpin H1 and H2 were heated to 95℃ for 5 min and cooled down to room temperature slowly.” This statement is confusing: both H1 and H2 on electrode or H1 and H2 were annealed in its own tubes?

5)      What buffer was used at hybridization and in analysis? Is the thermal stability is enough to form stable complexes at the surface at certain buffer conditions?

6)      All abbreviations must be decipher

7)      MHC, ABS, EISa and etc. – the abbreviation is unclear what.

8)       Line 113 “DNA hybrid buffer to use further.” What does mean “hybrid”?

9)      Experimental section. What temperature was used for PAGE experiments? Terms “four steps silver dye” is not widely used. Reformulate it. Line 137 “developer” – what does it means?

10)  Line 185 “phosphate skeleton” – replace by phosphate residues (moieties) or ribose-phosphate backbone.

11)  Figure 4(b) I would recommend show the only figure with lg(C). Linear scale is non-informative.

12)  Line 65 “electrochemical signal that was proportional to ctDNA concentration could be measured”. Here this statement is not substantiated.

13)  Figure S1 – figure labels and labels in figures must be different. Signing (a) for subfigure and curve designation lead to the misunderstanding.

Author Response

Reviewer #3:

The article is devoted to the development of a highly sensitive electrochemical biosensor for DNA detection using additional target treatment with Exonuclease 1 and subsequent HCR to amplify the detection signal. In the paper showed very promising results. Extremely high sensitivity of detection 1800 molecules in sample (0.5 fM, 6 ul of sample) was shown in the study. Moreover, using the three sigma rule they are proposing the 7 aM concentration (25 molecules) detection limit. However, many issues need to be resolved before publication. The manuscript needs major revision and after corrections re-revised.

Major problems are following:

Q1): There are a number of publication on electrochemical DNA biosensor with HCR amplification (for example, https://doi.org/10.1016/j.bios.2018.04.046, https://doi.org/10.1155/2017/4571614, https://doi.org/10.1016/j.talanta.2017.11.036) and HCR-Ag optical (https://doi.org/10.1039/D0RA04202A) or electrochemical (https://doi.org/10.1016/S1388-2481(02)00542-8) sensors. The novelty of the paper should be more clearly indicated.

Response: We appreciate the reviewer for the valuable suggestion. This work takes into account the background signal problem caused by DNA metallization. Exo Ι is selected to assist the background suppression strategy because the removal of unreacted capture probes from the electrode surface resulted in a relative reduction in the background signal. Moreover, DNA metallization assists in reflecting the HCR amplification strategy. We have double-checked and revised the manuscript (please see Page 2, line number 50-54 in the revised manuscript).

Q2): In the manuscript, the authors propose to use the KRAS gene ctDNA as a target. Typical ctDNA size is 166 base pairs (doi: 10.1073/pnas.1500076112). The authors propose to use short 19 nt fragments of model single stranded DNA. There are a number of problems for the use of ctDNA. It should be discussed in the manuscript.

  • It is not clear whether the KRAS gene fragment is present as ctDNA in the blood?

Response: It is clear that KRAS gene fragment is present as ctDNA in the blood. Nucleic acids were isolated from patient sera using a Norgen Biotek kit catalog number 51000. PCR-based method can analyze ctDNA in serum collected from lung cancer patients (KRAS mutations) (DOI: 10.1021/jacs.6b05679).

(b) How to denature double-stranded DNA and remove the second strand to prevent renaturation?

Response: The methods such as DNA clutch probes (DCPs) (DOI: 10.1021/jacs.6b05679, DOI: 10.1038/nchem.2270) and magnetic bead adsorption (DOI: 10.1002/adma.201801690) can denature double-stranded DNA and remove the second strand to prevent renaturation. The first method reported the electrochemical detection of mutated ctDNA in samples collected from cancer patients. By developing a strategy relying on the use of DNA clutch probes (DCPs) that render specific sequences of ctDNA accessible, the presence of mutated ctDNA could be read out. DCPs prevent the re-association of denatured DNA strands: they make one of the two strands of a dsDNA accessible for hybridization to a probe, and they also deactivate other closely related sequencesthe  in solution. DCPs ensure thereby that only mutated sequences associate with chip-based sensors detecting hybridization events. The second method designed magnetic nanopliers to remove complementary strands of the target mutant-type KRAS (C-MT-KRAS), preventing them rehybridize to mutant KRAS. Magnetic nanopliers are functionalized with peptide nucleic acid probes (PNA3) complementary to C-MT-KRAS. After the hybridization of C-MT-KRAS and magnetic separation, C-MT-KRAS were removed. In our systems, we used commercial target single-stranded DNA, which does not require double-stranded denaturation.

(c) What is the position of the selected target fragment in ctDNA and the location of the mutation? This is important for Exo 1 treatments.

Response: Exo Ι showed exonuclease activity to degrade single-stranded DNA from 3' to 5' while no activity on double-stranded DNA (DOI: 10.1016/j.bios.2018.09.081). There is no specific function on individual mutation sites.

Q3): How the sequence of target in 134A allele was selected? What is the origin of the choice of length and sequence for the target, the mismatches, and all the components of the system under study?

Response: The target (134A allele) was selected from the article (DOI: 10.1038/nchem.2270). The mismatches were selected from the article (DOI: 10.1021/acs.analchem.1c04037). The capture probe was designed according to the target. The initiator was designed according to the target and selected from article (DOI: 10.7150/thno.49047, 10.1016/j.bios.2013.03.020, 10.1016/j.jhazmat.2021.126223). The Hairpin H1 and H2 were selected from article (DOI: 10.7150/thno.49047, 10.1016/j.bios.2013.03.020, 10.1016/j.jhazmat.2021.126223).

Q4): section 3.3. a significant increase in current was observed after HCR. A reference experiment without a template but in the presence of all other components is required to validate the signal specifity.

Response: We appreciate the reviewer for the valuable suggestion. When the target is not present and all the other components are present, the capture probe is completely cut by the enzyme without a current signal. Under the same target concentration condition (1 nM), the current without HCR amplification was about 1.2 µA while the current with HCR treatment was about 4.9 µA. The current intensity was greatly increased after the HCR, where about 5 times higher than that without HCR, confirming the amplification strategy is absolutely feasible. We have double-checked and revised the manuscript (please see Page 6, line number 224-225 and Figure 3a in the revised manuscript).

Q5): The “Then, 6 µL MCH (1 mM) was dripped onto the Au electrode and incubated for 20 min to fill the blank area and prevent unspecific absorption.” The prevention of non-specific absorption was not shown.

Response: MCH contains mercapto groups that can form Au-S bonds with gold and avoid non-specific sites. MCH is for prevention. Successful modification of MCH was generally confirmed using CV and EIS (Please see Figure S1 and other references DOI:10.1007/s00216-021-03733-6, DOI: 10.1021/acssensors.9b00237).

Q6): “Taking the unique electrochemical advantage of Ag-AgCl solid-phase transformation [27], electrochemical signal that was proportional to ctDNA concentration could be measured.” I cannot fully agree with this statement. The signal is proportional not only to the ctDNA concentration, but also to the length of the amplified HCR chain.

Response: The signal is proportional to the ctDNA concentration and the length of the amplified HCR chain indeed. But the length of the amplified HCR chain was brought by the ctDNA concentration. The number of amplified HCR chains was proportional to the ctDNA concentration. So, we thought the electrochemical signal brought by Ag-AgCl solid-phase transformation was proportional to ctDNA concentration. The signal proportional to the target concentration also showed in this article (DOI: 10.1002/adfm.201303818).

Q7): The origin of the extremely high selectivity (single mismatch) presented in Figure 4(c) needs to be discussed.

Response: The single mismatch sequence was selected from the article (DOI: 10.1021/acs.analchem.1c04037).

Q8): Additional information on testing of real samples must be provided (section 3.6).

  • “The complex matrix solution was prepared by diluting human serum (from health volunteers) 100 times.” This sentence is not clear.

Response: We appreciate the reviewer for the valuable suggestion. Human serum from normal volunteers was drawn by the correct and reasonable blood drawing technique. The complex matrix solution was yielded by diluting human serum 100 times using buffer. We have double-checked and revised the manuscript (please see the Page 7, line number 269-271 in the revised manuscript).

(b) This Table S2 is not clear.

Response: We have supplemented the relevant information in Table S2, please see the revised supporting information.

(c) How many samples/donors was analyzed? What volume of serum was prepared and taken for analysis?

Response: In our experiment, 1 mL of serum was prepared from one healthy donor. And 1 µL of serum for dilution to analysis.

(d) The concentration of ctDNA was not measured by alternative approach in the serum.

Response: In our system, we defaulted to the inexistence of target ctDNA in the serum of healthy volunteers, consistent with the article (DOI: 10.1021/acs.analchem.1c04037).

(e) What does mean recovery experiment?

Response: The recovery experiment means that quantitative target is added to the blank sample without/with target, and the recovery rate is obtained under a comparison of tested and theoretical values according to the test procedure.

(f) Do you have permeation from ethical committee for experiments with volunteers and serum?

Response: We appreciate the reviewer for the valuable suggestion. We have permeation from the ethical committee for experiments with volunteers and serum. We have double-checked and revised the manuscript (please see Page 7, line number 268-269 in the revised manuscript).

Q9): “The recovery experiment realized in human serum sample provided not only a brand-new platform for early diagnosis of cancer, but also huge potential for the future analysis of other clinical samples.” To make this conclusion authors should analyze typical concentration of ctDNA at different cancer stages and compare them with the data obtained.

Response: We have modified this conclusion in our article (please see Page 8, line number 284-285 in the revised manuscript).

Minor issues:

Q10): In general, it is not correct to compare concentration limits of assays at different volumes used in the assay (Table S1). The molar quantity of samples is more appropriate.

Response: The aim of this work is to develop a universal assay for the detection of KRAS. The molar quantity of samples is a better way to compare a variety of methods. However, the comparison of the detection range and detection limits is more intuitive, and is often used in the article (DOI: 10.1021/acs.analchem.2c01041, 10.1039/d1tb02545g, 10.1021/acsami.7b15434).

Q11): Authors uses MCH (is it mercaptohexanol?) “to block the left sites on the electrode surface, thereby reducing the false positive signals”. Why don't you use a mixture of MCH with oligonucleotides to control the surface density of capture probes and fulfilling the surface?

Response: MCH refers to 6-mercapto-1-hexanol, which contains -SH groups that can form Au-S bonds with gold electrode to avoid non-specific absorption. If we use a mixture of MCH with oligonucleotides, it will compete with the binding site with capture, further reducing the modification efficiency of the capture probe on the electrode surface.

Q12): Authors claim that they “incubated for 1.5 h at room temperature to form self-assembled monolayers.” How did you confirm the monolayer formation?

Response: In our work, the density of the electrode surface-confined DNA probes, that is, the density of the capture probe can be estimated electrochemically via chronocoulometry (CC), which is similar to this article (DOI: 10.1016/j.bios.2016.02.011). According to the obtained results, we speculate that monolayers can be formed.

Q13): “After the electrode was rinsed with ultrapure water, hairpin H1 and H2 were heated to 95℃ for 5 min and cooled down to room temperature slowly.” This statement is confusing: both H1 and H2 on electrode or H1 and H2 were annealed in its own tubes?

Response: We appreciate the reviewer for the valuable suggestion. H1 and H2 were annealed in its own tubes by heated to 95℃ for 5 min and cooled down to room temperature slowly. Afterward, we mixed them to 1.5 µM containing hairpin H1 and H2 by buffer (50 mM Mg (NO3)2·6 H2O, 100 mM NaNO3, 20 mM HEPES, pH 7.4). Then, the mixed solution (1.5 µM) was applied to the electrode. We have double-checked and revised the manuscript (please see the Page 3, line number 116 in the revised manuscript).

Q14): What buffer was used at hybridization and in analysis? Is the thermal stability enough to form stable complexes at the surface at certain buffer conditions?

Response: Buffer (50 mM Mg (NO3)2·6 H2O, 100 mM NaNO3, 20 mM HEPES, pH 7.4) was used at hybridization and in analysis. The thermal stability is enough to form stable complexes at the surface at certain buffer conditions and it has been reported to be used (DOI: 10.1002/adfm.201303818). We have double-checked and revised the manuscript (please see the Page 3, line number 120-122 in the revised manuscript).

Q15): All abbreviations must be deciphered.

Response: We appreciate the reviewer for the valuable suggestion. All abbreviations have been deciphered in the article. We have double-checked and revised the manuscript (please see the Page 8, line number 300-305 in the revised manuscript).

Q16): MCH, APS, EIS and etc. – the abbreviation is unclear what.

Response: We appreciate the reviewer for the valuable suggestion. The abbreviation of MCH, APS, EIS and etc is clear (please see the MCH (Page 2, line number 56), APS (Page 4, line number 140-141, EIS (Page 3, line number 94-95), HCR (Page 1, line number 12), ctDNA (Page 1, line number 26), E-DNA (Page 1, line number 37), TEMED (Page 3, line number 83)) in the revised manuscript). All abbreviations have been deciphered in the article. We have double-checked and revised the manuscript (please see the Page 8, line number 300-305 in the revised manuscript).

Q17): Line 113 “DNA hybrid buffer to use further.” What does mean “hybrid”?

Response: “Hybrid” means the buffer contains Mg2+ that favors mutual binding between complementary single-strand nucleic acids.

Q18): Experimental section. What temperature was used for PAGE experiments? Terms “four steps silver dye” is not widely used. Reformulate it. Line 137 “developer” – what does it means?

Response: Room temperature was used for PAGE experiments. “Four steps silver dye” is reformulated with silver-stained method. “Developer” means the solution containing Na2CO3 and HCHO that reduces silver ions. We have double-checked and revised the manuscript (please see the Page 4, line number 142 in the revised manuscript).

Q19): Line 185 “phosphate skeleton” – replace by phosphate residues (moieties) or ribose-phosphate backbone.

Response: We appreciate the reviewer for the valuable suggestion. We have double-checked and revised the manuscript (please see the Page 5, line number 189 in the revised manuscript).

Q20): Figure 4(b) I would recommend show the only figure with lg(C). Linear scale is non-informative.

Response: We appreciate the reviewer for the valuable suggestion. We have double-checked and revised the manuscript (please see the Figure 4b in the revised manuscript).

Q21): Line 65 “electrochemical signal that was proportional to ctDNA concentration could be measured”. Here this statement is not substantiated.

Response: We appreciate the reviewer for the valuable suggestion. The signal is proportional the ctDNA concentration and the length of the amplified HCR chain indeed. But the length of the amplified HCR chain was brought by the ctDNA concentration. The number of amplified HCR chains was proportional to the ctDNA concentration. So, we thought the electrochemical signal brought by Ag-AgCl solid-phase transformation was proportional to ctDNA concentration. The signal proportional to the target concentration also showed in this article (DOI: 10.1002/adfm.201303818).

Q22): Figure S1 – figure labels and labels in figures must be different. Signing (a) for subfigure and curve designation lead to the misunderstanding.

Response: We appreciate the reviewer for the valuable suggestion. We have double-checked and revised the supporting information (please see the Pages 1, 2 and Figure S1 in the revised supporting information).

Round 2

Reviewer 3 Report

After revision, the manuscript became more suitable. However, minor corrections to the main text and the SI should be made before publication.

The authors in the answers gives a lot of information that is important to reader. Please add section in SI on selection of component of the biosensor with appropriate references (like in the answers to reviewer).

The answer on the following question (numbering in the answers to revision) should be added in the main text before publication.

Q7

The ability to distinguish between single mismatches is not a trivial task. Most techniques require sophisticated turning to make this possible. Please add to the main text a discussion of the origin of the observed extremely high selectivity with excellent sensitivity.

Q8-Q9

Information about ctDNA (length, method of obtaining) used for analysis in samples in serum should be added to the main text. Is this the target oligonucleotide or full-size double stranded DNA? If it is a short fragment of an oligonucleotide, you cannot call it ctDNA, as this is misleading to the reader.

Q10

Please add information about the volumes used for the analysis so that readers can adequately compare the sensitivity of different approaches.

Q12

The speculation is not confirmation. Please correct the main text.

Q17

Most probably you mean “DNA hybridization buffer”. Please, correct.

Author Response

Manuscript ID: biosensors-1965589

Title: "Target Recognition-and HCR Amplification-Induced In Situ Electrochemical Signal Probe Synthesis Strategy for Trace ctDNA Analysis"

Authors: Aiting Cai, Luxia Yang, Xiaoxia Kang, Jinxia Liu*, Feng Wang, Haiwei Ji, Qi Wang, Mingmin Wu, Guo Li, Xiaobo Zhou, Yuling Qin, Li Wu*

Dear Editors and Reviewers:

Thank you for your letter and for the reviewers’ comments concerning our manuscript entitled “Target Recognition-and HCR Amplification-Induced In Situ Electrochemical Signal Probe Synthesis Strategy for Trace ctDNA Analysis” (ID: biosensors-1965589). Those comments are all valuable and very helpful for revising and improving our paper, as well as the important guiding significance to our researches. We have studied comments carefully and have made correction which we hope meet with approval. Revised portion are marked with green background in the paper. Comments are written in blue and responses of the authors in black. The main corrections in the paper and the responds to the reviewers’ comments are as following:

Reviewer(s)' Comments to Author:

Reviewer #3:

After revision, the manuscript became more suitable. However, minor corrections to the main text and the SI should be made before publication.

The authors in the answers gives a lot of information that is important to reader. Please add section in SI on selection of component of the biosensor with appropriate references (like in the answers to reviewer).The answer on the following question (numbering in the answers to revision) should be added in the main text before publication.

Response: We appreciate the reviewer for the valuable suggestion. According to your advice, we have added the appropriate references in the revised supporting information on the selection of components of the biosensor (Please see the Page 9-10 in the revised supporting information).

Q7

The ability to distinguish between single mismatches is not a trivial task. Most techniques require sophisticated turning to make this possible. Please add to the main text a discussion of the origin of the observed extremely high selectivity with excellent sensitivity.

Response: We quite agree with you that most techniques require sophisticated task to distinguish between single mismatches. However, for electrochemical biosensor, it exhibited natural advantages for ctDNA detection due to their high specificity and sensitivity. A series of electrochemical biosensors have been established, which can effectively distinguish between single mismatches with simple design (DOI: 10.1021/acs.analchem.1c04037; 10.1016/j.cclet.2020.06.030; 10.1021/acsnano.1c11582; 10.1016/j.bios.2020.112821). In the system presented here, an electrochemical-DNA sensor was constructed by using DNA metallization strategy, as well as HCR and Exonuclease Ι (Exo Ι) cleavage technique, which exhibited excellent performance to distinguish a single base mutation as the individual molecular recognition differences can be converted into large different electrochemical signals through the above amplification strategies. The discussion of the origin of the observed extremely high selectivity with excellent sensitivity had been added in revised manuscript (please see the Page 7, line number 254 -257 in the revised manuscript).

Q8-Q9

Information about ctDNA (length, method of obtaining) used for analysis in samples in serum should be added to the main text. Is this the target oligonucleotide or full-size double-stranded DNA? If it is a short fragment of an oligonucleotide, you cannot call it ctDNA, as this is misleading to the reader.

Response: We appreciate the reviewer for the valuable suggestion. The length of the target used for analysis in samples in serum is 19 nt and it was obtained from a commercial target single chain (Sangon Biotech Co., Ltd. (Shanghai, China)). It is a short fragment of an oligonucleotide. We have double-checked and revised the manuscript (please see the Page 8, line number 274-275 in the revised manuscript).

Q10

Please add information about the volumes used for the analysis so that readers can adequately compare the sensitivity of different approaches.

Response: We appreciate the reviewer for the valuable suggestion. According to your suggestion, Information of the volumes used for the analysis had been added in revised manuscript. Please see the revised supporting information (please see the Page 8 in the revised supporting information).

Q12

The speculation is not confirmation. Please correct the main text.

Response: We have modified this conclusion in our article. We have double-checked and revised the manuscript (please see the Page 3, line number 107 in the revised manuscript).

Q17

Most probably you mean “DNA hybridization buffer”. Please, correct.

Response: We appreciate the reviewer for the valuable suggestion. We have double-checked and revised the manuscript (please see the Page 3, line number 117 and 120 in the revised manuscript).

We thank the reviewers again for their thoughtful comments. We believe that incorporating the suggestions from the reviewers has resulted in a greatly improved manuscript, and we look forward to your final decision on its suitability for publication in Biosensors.

Sincerely yours,

Li Wu

School of Public Health,

Nantong University, No. 9, Seyuan Road,

Nantong City, Jiangsu Province, Postal code: 226019

Tel: +86-513-55003046

E-mail: wuli8686@ntu.edu.cn